# Concerns and adjustments: How the Portuguese population met COVID-19

**Sara Valente de Almeida**[1][☉]*, **Eduardo Costa**[1][☉], **Francisca Vargas Lopes**[2][‡], **João Vasco Santos**[3,4,5][‡], **Pedro Pita Barros**[1][‡]

**1** Nova School of Business and Economics, Universidade NOVA de Lisboa, Campus de Carcavelos, Carcavelos, Portugal, **2** Department of Public Health, Erasmus Medical Center, Rotterdam, Netherlands, **3** MEDCIDS—Department of Community Medicine, Information and Health Decision Sciences, Faculty of Medicine, University of Porto, Porto, Portugal, **4** CINTESIS—Centre for Health Technology and Services Research, Porto, Portugal, **5** Public Health Unit, ACES Grande Porto VIII - Espinho/Gaia, ARS Norte, Portugal

☉ These authors contributed equally to this work.
‡ These authors also contributed equally to this work.
* sara.v.almeida.2015@novasbe.pt

**Data Availability Statement:** All relevant data are within the manuscript and its Supporting Information files.

**Funding:** This work was supported by Fundação para a Ciência e a Tecnologia [PhD studentship:

## Abstract

### Background

The COVID-19 pandemic has led to disruptive changes worldwide, with different implications across countries. The evolution of citizens' concerns and behaviours over time is a central piece to support public policies.

### Objective

To unveil perceptions and behaviours of the Portuguese population regarding social and economic impacts of the COVID-19 pandemic, allowing for more informed public policies.

### Methods

Online panel survey distributed in three waves between March 13[th] and May 6[th] 2020. Data collected from a non-representative sample of 7,448 respondents includes socio-demographic characteristics and self-reported measures on levels of concern and behaviours related to COVID-19. We performed descriptive analysis and probit regressions to understand relationships between the different variables.

### Results

Most participants (85%) report being at least very concerned with the consequences of the COVID-19 pandemic and social isolation reached a high level of adherence during the state of emergency. Around 36% of the sample anticipated consumption decisions, stockpiling ahead of the state of emergency declaration. Medical appointments suffered severe consequences, being re-rescheduled or cancelled. We find important variation in concerns with the economic impact across activity sectors.

BD128545/2017], Fundação para a Ciência e a
Tecnologia (UID/ECO/00124/2019, UIDB/00124/
2020 and Social Sciences DataLab, PINFRA/22209/
2016), POR Lisboa and POR Norte (Social
Sciences DataLab, PINFRA/22209/2016).

**Competing interests:** The authors have declared
that no competing interests exist.

## Conclusion

We show that high level of concern and behaviour adaptation in our sample preceded the
implementation of lockdown measures in Portugal around mid-March. One month later, a
large share of individuals had suffered disruption in their routine health care and negative
impacts in their financial status.

## 1 Introduction

At the end of 2019, the World Health Organisation (WHO) was informed by the Chinese
Authorities of an unknown cause pneumonia outbreak, in the city of Wuhan ([1] WHO
(2020)). The aetiological agent would soon be identified as a novel coronavirus, and later
designated as SARS-CoV-2. The corresponding disease was named COVID-19 ([2] WHO
(2020)). During January and February 2020, COVID-19 cases were increasingly confirmed in
countries outside China. On the $11^{th}$ of March, concerns over the alarming levels of disease
spread and severity led the WHO to declare the outbreak as a pandemic ([3] WHO (2020)). At
this point, Italy was the second most affected country in the world ([4] WHO (2020)) and
numbers in Europe were quickly increasing ([5] WHO (2020)). Soon other European coun-
tries including Spain, France, and the United Kingdom (UK) would see their incidence of
COVID-19 and associated mortality rise considerably.

The first confirmed case of COVID-19 in Portugal was identified on the $2^{nd}$ of March of
2020 ([6] DGS (2020)). Several national public health recommendations had already been
implemented at that time, restricting mass gatherings and public events or targeting travellers
returning from areas with presumed community transmission of COVID-19 ([6] DGS (2020)
and [7] DGS (2020)). Following educational facilities closure, the state of emergency was
declared by the Portuguese Government on the $18^{th}$ of March. It imposed the closure of non-
essential services and the duty of staying at home, except for strictly necessary activities (e.g.
buying food and medicines, working if not possible to do it remotely, or exercising individu-
ally). The state of emergency remained in place until the $2^{nd}$ of May ([8] DRE (2020)).

Similar measures have been adopted across the globe, with unprecedented impact on socie-
ties ([9] Baker et al. (2020), [10] Fernandes (2020), [11] Karabag (2020)). Most contingency
measures implemented by countries depended on individual voluntary commitment to be
effective in controlling the disease spread (WHO (2020)). Such a commitment can be affected
by messages on social media, political communication and institutional trust ([12] Biswas
(2016), [13] Verbeke and Viaene (2000), [14] Schmälzle et al. (2015), [15] Choi, Bull and Reed
(2020) and [16] Sadorsky (2012)).

While great uncertainty about the future remained, families were locked down at home,
collecting information that shaped their perceptions and deciding when and how to fulfil their
basic needs. Analysing people's perceptions of events and how those translated into decisions
and actions is of utmost importance to understand the effectiveness and collateral damages of
the pandemic-related measures. Do people grow more worried as the number of confirmed
cases increases? And do social isolation behaviours and consumption patterns change with
time and due to pandemic-related concerns? The monitoring of individuals' awareness, risk
perceptions, preventive behaviours, and trust is recommended by WHO to inform the pan-
demic outbreak response ([17] WHO (2020)). As in Portugal, several countries have been
implementing surveys with the above-mentioned objective, with results quickly becoming
available from across the world ([18] Atchison (2020), [19] Fetzer (2020), [20] Geldsetzer

(2020), [21] McFadden (2020), [22] Azlan (2020), [23] Meier (2020), [24] Smith (2020), [25] Trueblood (2020), [26] Seale (2020), [27] Sabat (2020), [28] Nazar (2020), [29] Peres (2020), [30] Bezerra (2020), [31] Wolf (2020) and [32] Nelson (2020)).

Inspired by the work of [33] Binder (2020) we developed an online panel survey to track the concerns and adjustments of the Portuguese population regarding the economic and social impact of the COVID-19 pandemic. The survey ran every 2 to 3 weeks and was launched very early in the course of the Portuguese outbreak, before most measures were implemented. We collected data on how people accessed information, perceived and reacted to the pandemic, and their expectations. Additionally, we measured changes in essential activities such as grocery shopping, working habits, and routine health care. We performed descriptive analyses and probit regressions to understand relationships between the different variables. Our objective was to analyse and quickly make available results of each wave, in order to inform society and aid in the design of interventions, policies, and communication strategies to tackle the subsequent stages of the pandemic ([17] WHO (2020)). The current paper adds to information made available through reports and media briefs. Knowing the evolution of Portuguese citizens' concerns and behaviours over time is central to understanding the outcome of the pandemic in the country. This knowledge is of relevance to inform future public health emergencies and more broadly to give insight about the Portuguese society in the context of crisis.

## 2 Data and methods

We prepared and distributed an online survey through social networks (WhatsApp, Facebook, Twitter and email) ([34] Guille et al. (2013)). The distribution strategy in social media was the same for the three waves. For the third wave we also emailed professional boards, hospitals, and patient associations, asking them to disseminate the survey amongst their associates. Looking at the socio-demographic characteristics of the respondents, we believe that this additional outreach strategy did not influence sample selection. Our study is cross-sectional, repeatedly collecting data from different participants to assess the status quo over time ([35] WHO) (for the respondents willing to participate in subsequent waves we added a longitudinal component to the survey, by introducing a self-generated code. Panel observations were dropped for this analysis (137 for the second wave and 150 for the third)). In addition to the self-reported data related to the pandemic, we collected individual-level information on socio-demographic characteristics such as age, gender, residence, education, income, occupation activity, and household composition (Table 1). Before voluntary participation, all participants were informed that their responses would be used for a scientific study conducted by Nova School of Business and Economics, and provided consent (by accepting a consent statement) for building a unique individual and anonymous identifier. The survey received ethical approval on March 13[th] by the Installing Committee of Nova School of Business and Economics' Ethics Commission (*Commissão Instaladora do Comité de Ética*).

In this paper we describe the first three waves of our survey, launched on the 13[th] and 27[th] of March, and on the 17[th] of April 2020. Responses collected until May 6[th] were included in the analysis. As recommended in such dynamic situations, we have added/removed questions to adapt our survey to the ongoing phase of the pandemic ([17] WHO (2020)). The first-wave instrumented had 33 questions. Additional questions about access to health care services and concerns regarding the financial situation were added to the second and third-wave questionnaires. Table 1 highlights the main characteristics of each wave.

Our repeated cross-sectional dataset includes the answers of 7,448 respondents, which we analysed as a pooled and time series sample. We performed descriptive analyses and

**Table 1. Key information of survey waves.**

|  | **Wave 1** | **Wave 2** | **Wave 3** |
|---|---|---|---|
| Start date | March 13 | Mach 27 | April 17 |
| End date | March 26 | April 16 | May 6 |
| Number of questions | 33 | 49 | 47 |
| Respondents | 5,460 | 969 | 1,019 |
| Question topic: |  |  |  |
| Socio-demographic characterization | x | x | x |
| Symptoms and isolation measures | x | x | x |
| Concerns with the pandemic | x | x | x |
| Stockpiling goods | x | x | x |
| Search for information | x | x | x |
| Quality and reliability of information |  | x |  |
| Economic impact (current and forecast) |  | x | x |
| Capacity and disruption of health services |  | x | x |
| Expected compliance and agreement with isolation measures |  |  | x |

Note: Respondents were given the option not to answer particular questions. The number of respondents does not include participants who answered to any previous waves (as identified through a self-generated code) [42].

conducted probit models to study the association between perceptions/expectations and behaviours captured by our survey.

The probit model allows comparing outcomes between subjects included and not included in a given group (T = 1 and T = 0, respectively), which implies that the dependent variable is always binary. The predicted outcome is thus the estimated probability of belonging to the group or having a given preference. For example, we use a probit model to estimate the probability of being in social isolation. For each sampled subject in isolation (or not) we have an estimated propensity score, $\hat{P}(X|T=1) = \hat{P}(X)$. With the probit model, the conditional maximum likelihood estimator (MLE) finds a conditional density that depends only on observable data and the parameter $\beta$. To interpret the coefficients from a probit estimation we compute the marginal effects, which give a more direct interpretation than the output coefficients from the model. Marginal effects show the impact of a one-unit change in each explanatory variable on the outcome variable.

Additionally, for the particular case of the concerns and social isolation analyses, due to the nature of the variables and the particularly strong correlation between them, we estimate a Structural Equation Model (SEM). These models consist of a system of simultaneous equations that fit linear models for continuous responses. By estimating multiple and interrelated dependence in a single analysis, the model allows unobserved components in the equations and correlation between equations. As such, although in this framework it is not possible to completely control for all endogeneity issues, the model is able to provide valid and unbiased results controlling for highly correlated variables, such as general and economic concerns, teleworking, and being in social isolation (please refer to [36] Wooldridge (2002) for a more detailed description of both the Probit and the SEM models).

In all models we control for gender, age group, education level, and region. Age is considered in seven different age groups with five-years interval between thresholds, starting at below 18 years old and ending at more than 80 years old. Education is included in the model as a binary variable indicating whether the subject completed a higher education level. The survey was conducted using Qualtrics$^{XM}$ and the analysis with the statistical software Stata 14.

The goal of this paper is to provide insight on the most important findings from the state of emergency period. Despite the vast range of information collected, we focused our analysis on four main dimensions: social isolation and concern, stockpiling and consumption patterns, health services performance, and economic consequence (topics not studied in this paper include: symptoms, search for information, and quality and reliability of information). These dimensions were selected based on data completeness, meaningful results, and relevance for policy design. Detailed statistics for the questions in each dimension are presented in the appendix.

## 3 Results

### 3.1 Sample characteristics

Respondents in our sample were mostly from Lisbon (59%), followed by the North (18%) and Centre (14%) of the country. When compared to the national average there is an over-representation of women (73%) and highly educated individuals (78% attended college), while elderly and single household individuals are under-represented (Table 2).

About one fifth of the individuals reported living with members above 65 years old (21%) or with health professionals (20%). The most commonly reported professional occupations were in the areas of health (17%), economics, management and finance (14%), and education (13%) (Table 3).

One important and noteworthy caveat of the study is that the convenience sample used is not representative of Portuguese citizens at large, limiting the generalization of current findings (the discussion section of the paper includes further details on this caveat).

### 3.2 Social isolation and concern

At the beginning of the survey (March 13th) 18% of the respondents reported having adhered to social isolation (in this context, social isolation refers to staying at home, but not necessarily without contact with the household members or for COVID-19 related symptoms—during the state of emergency citizens were required to stay at home except for essential activities such as buying groceries and medicines, going to work if remote working was not possible, and walking outside/exercising individually for a limited period of time). This proportion rose over time, notably after March 18th, when the state of emergency was declared (Fig 1). On April 6th, 100% of the respondents claimed to have adopted social isolation measures. A temporary drop in isolation is observed around the Easter weekend and a more pronounced decline can be identified from the last week of April and thereafter, when the first news about the end of the state of emergency(May 2nd) started to circulate (available in: https://www. reuters.com/article/us-health-coronavirus-portugal/portugal-to-lift-coronavirus-state-of-emergency-from-may-3-idUSKCN22A282).

Most participants reported being at least very concerned with the consequences of the COVID-19 pandemic (85%, S1 Table), with the youngest group (under 25 years old) being the least concerned of all. The share of respondents reporting a high level of concern regarding the pandemic was relatively stable over time.

We find a positive correlation between being concerned with the impact of the pandemic on the Portuguese economy and complying with social isolation. The results for the marginal effects of the probit model show that respondents who adopted social isolation were 4 percentage points (p.p.) more likely to be concerned with the economic impact (statistically significant at 1% level). In turn, being worried about the economy increased the probability of adopting social isolation by around 10 p.p., with the same significance level.

**Table 2. Sample characteristics and national average.**

| Variable | Number | % | National Average |
|---|---|---|---|
| **Age** | | | |
| <18 years | 68 | 1% | 19% |
| 19 to 25 years | 768 | 10% | 5% |
| 26 to 32 years | 1,458 | 20% | 11% |
| 33 to 45 years | 2,651 | 36% | 14% |
| 46 to 64 years | 2,144 | 29% | 29% |
| 65 to 79 years | 344 | 5% | 15% |
| >80 years | 15 | 0% | 7% |
| **Gender** | | | |
| Male | 2,037 | 27% | 47% |
| Female | 5,407 | 73% | 53% |
| Other | 4 | 0% | 0% |
| **Region** | | | |
| North | 1,323 | 18% | 35% |
| Centre | 1,074 | 14% | 22% |
| Lisbon | 4,405 | 59% | 28% |
| Alentejo | 223 | 3% | 7% |
| Algarve | 318 | 4% | 4% |
| Azores Islands | 50 | 1% | 2% |
| Madeira Island | 55 | 1% | 2% |
| **Education** | | | |
| Elementary school | 16 | 0% | 38% |
| Middle school | 235 | 3% | 32% |
| High school | 1,380 | 19% | 16% |
| University | 5,790 | 78% | 14% |
| **Number of household members** | | | |
| One / just me | 732 | 10% | 23% |
| Two | 2,223 | 30% | 77% |
| Three | 2,069 | 28% | |
| Four | 1,740 | 23% | |
| Five | 190 | 3% | |
| More than five | 494 | 7% | |

Note: 7,448 valid answers were used for this table. Respondents were given the option not to answer particular questions. Proportions (%) were computed based on the number of answers to each question excluding respondents who opted not to answer.

Source for national averages: Portuguese Statistics Institute (2019 and 2011 census)

Contrarily, the association between the general concern with the pandemic and isolation is in the opposite direction, with respondents in social isolation being less likely to be worried by 2 p.p.. In addition, teleworking is also relevant to explain social isolation adherence, being associated with a 6 p.p. increase in the probability of adopting it. Being a woman, of an older age, and having a higher education level have positive and statistically significant associations with the dependent variables studied (Table 4).

Given that the Probit model estimation presents some unexpected results, to further investigate these relationships and control for potential endogeneity issues, we estimate the SEM for the same dependent variables. The system is composed of three equations: regressions A, B,

**Table 3. Additional sample characterization—Household, income, and occupation.**

| Variable | Number | % |
|---|---|---|
| **Number of household members over 65 years** | | |
| None | 5,895 | 79% |
| One / just me | 826 | 11% |
| Two | 586 | 8% |
| Three | 71 | 1% |
| Four | 41 | 1% |
| Five | 9 | 0% |
| More than five | 20 | 0% |
| **Health professional in the household** | | |
| At least one | 1,486 | 20% |
| None | 5,962 | 80% |
| **Household monthly disposable income** | | |
| <1100 euros | 1,206 | 18% |
| 1101-1500 euros | 1,175 | 18% |
| 1501-2000 euros | 1,494 | 23% |
| 2001-5000 euros | 2,404 | 36% |
| >5001 euros | 329 | 5% |
| **Professional occupation** | | |
| Health | 1,259 | 17% |
| Economics, management, and finance | 1,001 | 14% |
| Education | 948 | 13% |
| Retail | 437 | 6% |
| Industry and utilities | 387 | 5% |
| Informatics | 338 | 5% |
| Construction and real estate | 237 | 3% |
| Research | 195 | 3% |
| Social care activities | 144 | 2% |
| Entrepreneurship | 109 | 2% |
| Transport | 77 | 1% |
| Marketing and communication | 62 | 1% |
| Agriculture | 41 | 1% |
| Law | 38 | 1% |
| Does not work / other | 1,955 | 27% |

Note: 7,448 observations were used for this table. Respondents were given the option not to answer particular questions. Proportions (%) were computed based on the number of answers to each question excluding respondents who opted not to answer. No definition of health professional was provided in the survey, responses are based on the participants understanding of this professional occupation.

and C. Regression A estimates the likelihood of adhering to social isolation as a function of concerns with economy, general concern with the pandemic and teleworking. Since concern with economy and general concern are highly correlated, we then use regression B and C to estimate these two variables separately. Note that for the system to be identified we add new variables related to the outcome variables, but not between them. To regression B, on concern with economy, we add a variable that indicates feeling already (high or very high) negative financial impacts due to the pandemic; and to regression C we add following the news (frequently or very frequently) as an explanatory variable for general concern. For the whole

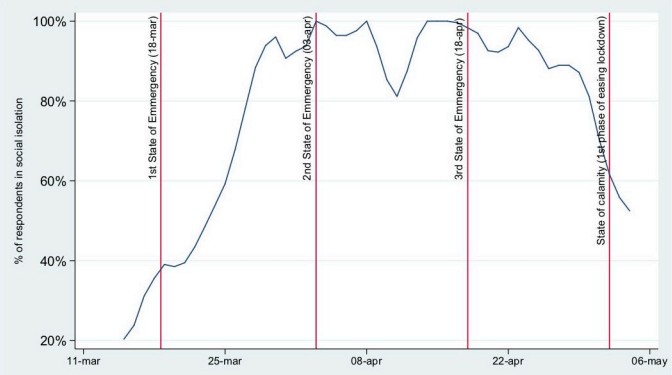

**Fig 1. Proportion of respondents with social isolation measures implemented by date (non-cumulative; 3-days moving average).** Note: Vertical lines mark the beginning, renewal and end of the state of emergency.

system, the first specification (I) includes only concern-related variables to explain the decision of practicing social isolation, while the second specification (II) includes working from home to the group of explanatory variables. We find similar evidence similar to that obtained with the Probit regressions for specification (I) Table 5. However, when teleworking is added to the system, the coefficients for concern with economy and general concern lose their significance. According to this evidence, adherence to social isolation depends on teleworking rather than on levels of concern with economy and the pandemic. Regarding the variables newly added to regressions B and C, both the perception of a current negative impact on financial status and following the news closely have a strong and positive impact on the respective concern variables.

**Table 4. Probit (margins) results for social isolation, concerns, and teleworking.**

| | Social Isolation | Concern with economy | General concern | Teleworking |
|---|---|---|---|---|
| Social Isolation | | 0.044*** | -0.017* | 0.105*** |
| | | (0.008) | (0.009) | (0.035) |
| Concern with economy | 0.098*** | | | |
| | (0.018) | | | |
| General concern | -0.033** | | | |
| | (0.017) | | | |
| Teleworking | 0.056*** | | | |
| | (0.018) | | | |
| Controls | | | | |
| - Gender (female) | + | . | + | . |
| - Age group (5 cat.) | + | + | + | - |
| - Higher education | + | + | . | + |
| - Residence area (Lisbon) | ✓ | ✓ | ✓ | ✓ |

*,**, *** indicate significance at 10%, 5%, and 1% level, respectively; (standard error).

Note: Concern with economy is a binary variable that indicates whether the respondent was concerned or very concerned with the pandemic's impact on the Portuguese economy; general concern is a binary variable that indicates whether the respondent was concerned or very concerned with the pandemic in Portugal; social isolation is a binary variable that indicates whether the respondent reported having adopted social isolation; finally, teleworking indicates whether the participant is working from home (1) or not (0). For the controls, the sign indicates the sign of the coefficient, if no sign is presented the coefficient was not statistically significant.

**Table 5. SEM results for social isolation, concerns and teleworking.**

|  | (I) | (II) |
|---|---|---|
| **A—Social Isolation** |  |  |
| Concern with economy | 0.095*** | 0.041 |
|  | (0.018) | (0.035) |
| General concern | -0.056** | 0.019 |
|  | (0.017) | (0.023) |
| Teleworking |  | 0.044* |
|  |  | (0.020) |
| Controls |  |  |
| - Gender (female) | + | . |
| - Age group (65+) | + | . |
| - Higher education | + | . |
| - Residence area (Lisbon) | - | - |
| **B—Concern with economy** |  |  |
| Negative present impact | 0.071*** | 0.053*** |
|  | (0.011) | (0.013) |
| Controls |  |  |
| - Gender (female) | . | . |
| - Age group (65+) | + | + |
| - Higher education | + | . |
| - Residence area (Lisbon) | . | + |
| **C—General concern** |  |  |
| Follow news closely | 0.327*** | 0.267*** |
|  | (0.028) | (0.039) |
| Controls |  |  |
| - Gender (female) | + | + |
| - Age group (65+) | + | + |
| - Higher education | + | - |
| - Residence area (Lisbon) | . | . |

*,**,*** indicate significance at 10%, 5%, and 1% level, respectively; (standard error).

Note: All dependent variables of interest are binary variables. Concern with economy indicates whether the respondent was concerned or very concerned with the pandemic's impact on the Portuguese economy; general concern indicates whether the respondent was concerned or very concerned with the pandemic in Portugal; social isolation indicates whether the respondent reported having adopted social isolation; negative present impact indicates whether the respondent reported to already feeling high or very high negative impact of the pandemic in the financial situation of the household; follow news closely indicates whether the respondent follows news frequently or very frequently; finally, teleworking indicates whether the participant is working from home or not at the time of the survey. For the controls, the sign indicates the sign of the coefficient, if no sign is presented the coefficient was not statistically significant. In this model, factor variables were modified into binary, such that Age group indicates subjects older than 65 years old and Residence Area indicates subjects living in Lisbon.

## 3.3 Stockpiling and consumption patterns

Around 36% of our sample reported stockpiling even before the state of emergency was declared (Fig 2), with a peak around the second week of March. This behaviour was more pronounced within young professionals (between 35 and 45 years old). Half of the second wave respondents (52%) (question introduced only from the second wave on). reported having difficulties buying at least one essential good (S2 Table).

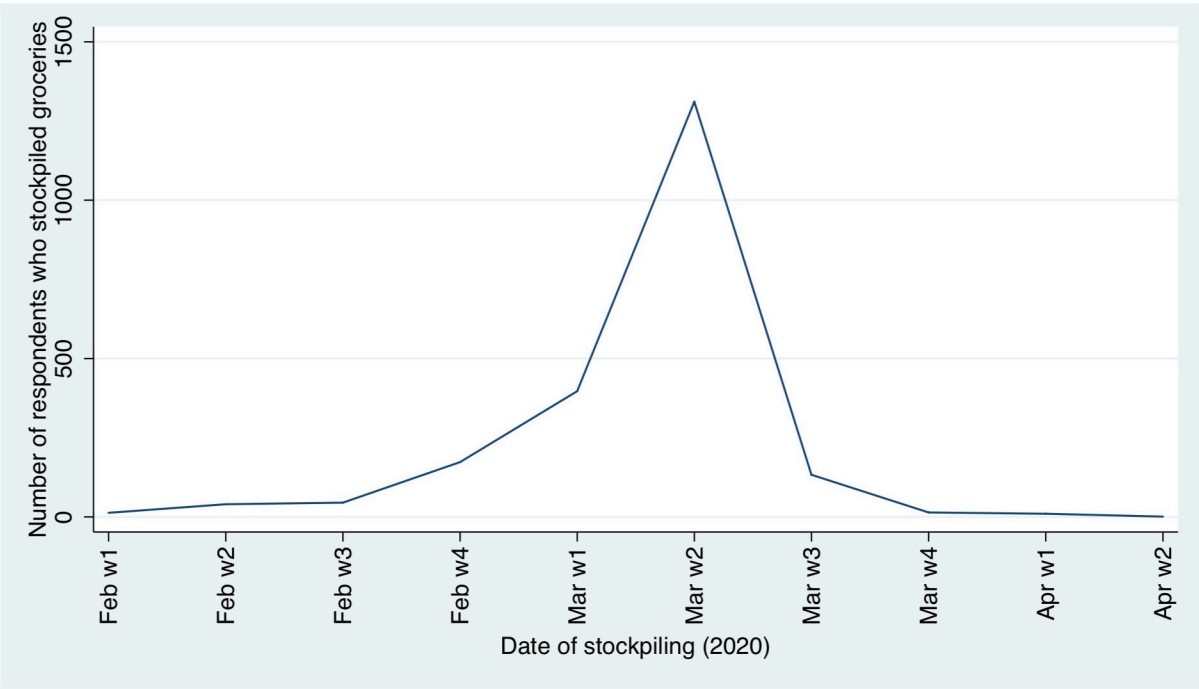

**Fig 2. Stockpiling of groceries and other goods.**

The marginal effect of being concerned with the pandemic on the probability of stockpiling is 13 p.p. (Table 6). In the opposite direction, respondents who stockpiled groceries were 7 p.p. more likely to report concerns with the pandemic (both significant at 1% level). Being younger and highly educated was associated with stockpiling.

## 3.4 Health services capacity

When asked about whether the health services capacity had changed over the previous month, most respondents of the second and third waves reported being unable to assess it (54%). Of

**Table 6. Probit (margins) results for stockpiling and general concern with the pandemic.**

|  | Stockpiling | General concern |
|---|---|---|
| General concern | 0.133*** |  |
|  | (0.009) |  |
| Stockpiling |  | 0.074*** |
|  |  | (0.0426) |
| Controls |  |  |
| - Gender (female) | . | + |
| - Age group (5 cat.) | - | + |
| - Higher education | + | - |
| - Residence area (Lisbon) | ✓ | ✓ |

*,**, *** indicate significance at 10%, 5%, and 1% level, respectively; (standard error).

Note: Stockpiling is a binary variable that indicates whether the respondent reported making extraordinary shopping decisions due to the pandemic; concern is a binary variable that indicates whether the respondent was concerned or very concerned with the pandemic in Portugal. For the controls, the sign indicates the sign of the coefficient, if missing the coefficient was not statistically significant.

**Table 7. Probit (margins) results for health appointment change and type of provider.**

| | Appoint. change (a) | Appoint. change (b) | Private service | Public service |
|---|---|---|---|---|
| Private service | 0.133*** | | | |
| | (0.033) | | | |
| Public service | | 0.004 | | |
| | | (0.035) | | |
| Appoint. change | | | 0.308*** | 0.002 |
| | | | (0.074) | (0.082) |
| Controls | | | | |
| - Gender (female) | . | . | . | . |
| - Age group (5 cat.) | . | . | . | . |
| - Higher education | . | . | + | - |
| - Residence area (Lisbon) | ✓ | ✓ | ✓ | ✓ |

*,**, *** indicate significance at 10%, 5%, and 1% level, respectively; (standard error).

Note: Appointment change is a binary variable that indicates whether the respondent had an appointment postponed/cancelled or not; private services is a binary variable that indicates whether the respondent had that appointment scheduled at a private health facility; and by correspondence, public services variable indicates whether the appointments were in a facility from the NHS (SNS). This regression includes only respondents who had an appointment scheduled and respondents could have appointments booked in both services simultaneously. For the controls, the sign indicates the sign of the coefficient, if missing the coefficient was not statistically significant. The question used for this regression was added only to wave 2 and thereafter, and thus this does not include information from the first wave.

those who were able (or willing) to quantify this, half considered that the health system capacity had remained stable (24% overall, S3 Table).

Despite these perceptions about the health services capacity, several patients experienced delays or changes in their planned medical appointments. About 44% of the respondents had at least one medical appointment planned for the coming months. Of those, 90% had a change in scheduling of at least one of their appointments. *Circa* 6% had their appointment changed from face-to-face to remote appointment and the remaining respondents had appointments postponed or cancelled. Most appointments postponed or cancelled were with private providers (56%), followed by the NHS (47%) (some respondents had multiple appointments in both the private sector and the NHS, thus the sum of appointments postponed is greater than 100%).

Table 7 shows the relationship between appointment postponed/cancelled and type of service. In this regression we include only subjects with scheduled appointments that might have been postponed/cancelled due to the pandemic or other. The regression results indicate that having an appointment scheduled in a private medical service increased the probability of having a cancellation or postponement by 13 p.p., significant at 1% level. For public services this probability is close to nil and not statistically significant. Gender, age, and education were not significantly associated with having an appointment changed.

### 3.5 Economic consequences

Almost two-thirds of the second and third wave respondents (64%) claimed they had already experienced negative financial impacts due to the pandemic at the time of answering the survey. These negative impacts were qualified as small or moderate for 58% of the individuals. When asked about financial impacts in the future (short-run), the proportion of individuals that believed their financial situation would be negatively impacted increased to 83% (S4 Table). We find variations in the expectations about the financial situation across different activity sectors, with tourism-related occupations reporting the worst predictions on their

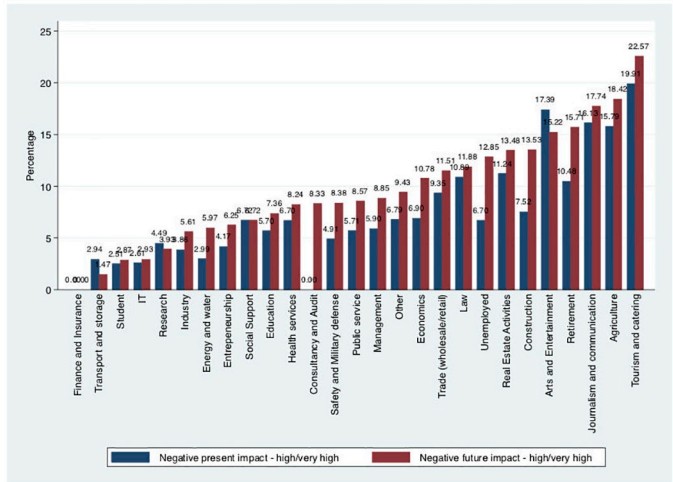

**Fig 3. Proportion of respondents feeling present and future negative financial impact of the pandemic.**

financial status (see Fig 3). "Arts and Entertainment", "Research", and "Transport and storage" were the only professional occupations with fewer individuals expecting less negative impact on their financial status in the future than present negative impact.

Table 8 reports the relationship between concerns with the economy and impact of the pandemic on the present and future financial situation. The probit model results show a positive relationship between concerns and perceptions, with individuals concerned being more likely to expect high or very high current (14 p.p.) or future (15 p.p.) negative financial impacts. On the other hand, reporting high or very high negative impacts in the current financial situation

**Table 8. Probit (margins) results for economic present and future impact and concern with economy.**

|  | Concern with economy | Negative present impact | Negative future impact |
|---|---|---|---|
| Negative present impact | 0.059*** |  |  |
|  | (0.017) |  |  |
| Negative future impact | 0.053*** |  |  |
|  | (0.016) |  |  |
| Concern with economy |  | 0.140*** | 0.147*** |
|  |  | (0.041) | (0.043) |
| Controls |  |  |  |
| - Gender (female) | + | . | . |
| - Age group (5 cat.) | . | - | - |
| - Higher education | - | . | + |
| - Residence area (Lisbon) | . | . | . |

*,**, *** indicate significance at 10%, 5%, and 1% level, respectively; (standard error).

Note: Negative present impact is a binary variable that indicates whether the respondent reported already feeling high or very high negative impact of the pandemic in the financial situation of the household; negative future impact is a binary variable that indicates whether the respondent reported foreseeing high or very high negative financial impacts for the household; concern with economy is a binary variable that indicates whether the respondent was concerned or very concerned with the pandemic impact on the Portuguese Economy. For the controls, the sign indicates the sign of the coefficient, if missing the coefficient was not statistically significant.

was associated with a 6 p.p. increased probability of being concerned with the economy, while expecting future impacts was 5 p.p..

## 4 Discussion

Our study comprehends the different phases of the Portuguese reaction to the COVID-19 pandemic, including information from the early phase. We present results of three survey waves that cover the period of 1.5 months during which a partial lockdown was in place in Portugal.

Several days before the state of emergency was declared (March 18$^{th}$ 2020), approximately one-third of the respondents reported being in social isolation, suggesting that part of the Portuguese population anticipated isolation measures. Similar anticipation had already been reported, for example, in Malaysia, where most of the residents were already taking preventive measures (e.g., avoiding crowds, proper hand hygiene, or wearing a face mask) the week before the movement control order by their Government ([22] Azlan et al. (2020)). This is in accordance with the public attention that sharply increased following the first COVID-19 case in Portugal (based on Google searches), with the peak being achieved when the state of emergency was first ordered ([37] Aksoy et al. (2020)).

Earlier studies on other pandemics have demonstrated that perceptions and behaviours often change over time ([38] Bults et al. (2015)), which we confirmed in our study. Additionally, perceptions and behaviours depend strongly on the phase of the pandemic faced in each country/region. In fact, a European survey between the 3$^{rd}$ and 4$^{th}$ week of March showed high levels of social isolation measures and personal protective behaviours in Italy compared to the Netherlands and Germany, possibly due to the more extensive lockdown measures and greater COVID-19 burden ([23] Meier et al. (2020)).

During a substantial part of the Portuguese state of emergency almost 100% of the respondents reported being in social isolation, contrasting with lower proportions described for the UK ([24] Smith et al. (2020)), though such high results might be due to our sample bias mentioned below. The easing of lockdown measures seems to have been anticipated by the Portuguese respondents as well, with a similarly high proportion no longer in isolation before the end of the state of emergency and transition to state of calamity (May 2$^{nd}$). Besides the link between socioeconomic factors and social isolation ([18] Atchison et al. (2020), [25] Trueblood et al. (2020), [39] Jay et al. (2020)), Seale et al. also described the association between the level of concern if self-isolated and the adoption of avoidance behaviours ([26] Seale et al. (2020)). Surprisingly, despite finding a positive association between the concern with COVID-19 pandemic in the Portuguese economy and the social isolation, the opposite (negative) association was found between the general concern level and social isolation. A possible explanation resides in the (possible) externality of being isolated in terms of general anxiety or concern levels. However, when considering SEM results, social isolation seems to depend on teleworking rather than on concern with the economy and the pandemic. Although this might have been in some way coerced by employers or the Government, a Belgian study described that those who were asked or forced to telecommute during this pandemic experienced less "perceived infectability" and "germ aversion" but higher levels of solidarity, i.e. "whether they will self-quarantine if they feel unwell" ([40] De Coninck et al. (2020)).

In our study, respondents seemed to be more concerned with the impact of the COVID-19 pandemic on the Portuguese economy (88% of the respondents were very or extremely concerned) in comparison with the pandemic's general impact (85% of the respondents were very or extremely concerned). In a survey conducted in seven European countries (i.e. Denmark, France, Germany, Italy, Portugal, the Netherlands and the UK) during the first half of April, Portuguese respondents were the most worried with all aspects studied, including the health

system, small businesses, recession, and death of a loved one, despite Portugal not being the country with the worst epidemiological situation, and having neither the most stringent measures nor the lowest economic support ([27] Sabat et al. (2020)). Levels of concern in our study were relatively stable over time, contrasting with a Polish survey that showed a large increase before and after the Polish declaration of the state of emergency. ([28] Nazar et al. (2020)).

Interestingly, Sabat et al. reported that 84% of respondents in Portugal stated that they worried "quite a bit or a lot" about the national health system becoming overloaded ([27] Sabat et al. (2020)), which is in line with our results, and follows the results by Peres et al., in which 50.1% of the Portuguese general population sample and 63.5% of the health professionals' sample stated that health services were poorly prepared or unprepared to deal with SARS-CoV-2 ([29] Peres et al. (2020)). Rearranging patient flows and adapting structures to deal with COVID-19 patients crowded out the provision of health services to the non-COVID patients, even though the maximum capacity made available for COVID-19 was not exhausted. In our study, from those willing to evaluate the quality of the health system since the beginning of the pandemic, one third stated that it had worsened, while approximately 15% stated that it had improved. For those respondents who had a medical appointment, almost 90% mentioned that the appointment was affected. A cancellation or postponement seemed to be more probable when scheduled with private health services rather than with the public services.

One of the reactions to increasing concern and adopting social isolation is the stockpiling of groceries and other goods. Stockpiling peaked in the second week of March, one week after the first COVID-19 case in the country and one week before implementing the state of emergency. The level of stockpiling fell with time, in line with what was shown in the Polish survey by Nazar et al. ([28] Nazar et al. (2020)). Around 36% of our sample anticipated consumption decisions, in contrast with a US survey reporting 74.7% of respondents stockpiling food and supplies ([32] Nelson et al. (2020)). In both surveys nearly a third of the respondents reported difficulties in finding some food supplies in supermarkets ([32] Nelson et al. (2020)).

Important findings from our second and third waves relate to the current and future financial impacts of this pandemic, with two-thirds of the participants already suffering some negative impact and the vast majority expecting a negative impact in the short run. Our figures show that families that felt negative impacts in their financial situation early on, or anticipated negative impacts in the future, reported higher concern levels, and vice-versa. With families being concerned and feeling negative impacts, their behaviours will follow. Understanding these interactions is essential to implement appropriate policies and communicate effectively, without undermining trust or promoting disbelief. Given that our sample seems to suffer from over-representation of urban and middle-high socioeconomic class females the magnitude of the (expected) financial impacts reported should be of great concern.

Our study suffers from selective participation, as do most online surveys with our dissemination methods. Therefore, our convenience sample is not representative of Portuguese citizens and lacks information on participation rates, limiting the generalization of current findings. For example, it has been reported that socioeconomic factors have implications in the perception or implementation of social isolation ([39] Jay et al. (2020), [30] Bezerra et al. (2020), [25] Trueblood et al. (2020)). Social isolation reached 100% of adherence during the state of emergency, while some activities did not close even during this period. Another aspect related to our study design is that surveys collecting self-reported data are generally subjected to limitations including honesty, introspective ability, and interpretation of the questions.

Well-being, social cohesion, economic stability, and resilience of individuals and communities are likely to be impacted by the pandemic and its restrictions. Collecting timely information to characterize this complex context allows the anticipation and mitigation of unintended scenarios and the implementation of better informed, situated, and accepted pandemic

response measures, which are likely to be more effective ([41] WHO (2004)). Our results can have a role in tailoring policies to better address concerns and adjustments of the Portuguese population, in meeting future phases of the current pandemic and future pandemics.

## Supporting information

**S1 Table. Survey questions: Social isolation and concern.**
(PDF)

**S2 Table. Survey questions: Stockpiling and consumption patterns.**
(PDF)

**S3 Table. Survey questions: Health Services capacity.**
(PDF)

**S4 Table. Survey questions: Economic consequences.**
(PDF)

**S1 File. Database used for calculations.**
(XLSX)

## Acknowledgments

We are very grateful for the helpful comments from all members of the Nova Health Economics & Management Knowledge Centre of Nova School of Business and Economics—Universidade Nova de Lisboa. We also thank two anonymous reviewers for helpful comments on earlier drafts of the manuscript. Any remaining errors are ours.

## Author Contributions

**Conceptualization:** Sara Valente de Almeida, Eduardo Costa, Francisca Vargas Lopes, João Vasco Santos, Pedro Pita Barros.

**Data curation:** Sara Valente de Almeida, Eduardo Costa.

**Formal analysis:** Sara Valente de Almeida, Eduardo Costa.

**Investigation:** Sara Valente de Almeida, Eduardo Costa, Francisca Vargas Lopes, João Vasco Santos.

**Methodology:** Sara Valente de Almeida, Eduardo Costa.

**Project administration:** Sara Valente de Almeida, Eduardo Costa.

**Software:** Sara Valente de Almeida, Eduardo Costa.

**Supervision:** Pedro Pita Barros.

**Validation:** Francisca Vargas Lopes, João Vasco Santos, Pedro Pita Barros.

**Visualization:** Sara Valente de Almeida, Eduardo Costa.

**Writing – original draft:** Sara Valente de Almeida, Eduardo Costa, Francisca Vargas Lopes, João Vasco Santos.

**Writing – review & editing:** Sara Valente de Almeida, Eduardo Costa, Francisca Vargas Lopes, João Vasco Santos, Pedro Pita Barros.

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
