## [Decision Letter · Decision Letter 0]

22 Jun 2020

PONE-D-20-15289

Concerns and Adjustments: How the Portuguese population met COVID-19

PLOS ONE

Dear Dr. Valente de Almeida,

Thank you for submitting your manuscript to PLOS ONE. After careful consideration, we feel that it has merit but does not fully meet PLOS ONE’s publication criteria as it currently stands. Therefore, we invite you to submit a revised version of the manuscript that addresses the points raised during the review process.

Both the referees have made suggestions on providing a little more details on some of the questions, and some new analysis (such as a 2x2 table on some metrics). I would agree with them. This is a simple descriptive paper, but is useful nevertheless as it tells us something about how sample respondents are viewing the pandemic.

We look forward to receiving your revised manuscript.

Kind regards,

Renuka Sane

Academic Editor

PLOS ONE

Journal Requirements:

Additional Editor Comments (if provided):

Reviewers' comments:

Reviewer's Responses to Questions

**Comments to the Author**

1. Is the manuscript technically sound, and do the data support the conclusions?

Reviewer #1: Yes

Reviewer #2: Yes

2. Has the statistical analysis been performed appropriately and rigorously? 

Reviewer #1: Yes

Reviewer #2: Yes

3. Have the authors made all data underlying the findings in their manuscript fully available?

Reviewer #1: Yes

Reviewer #2: Yes

4. Is the manuscript presented in an intelligible fashion and written in standard English?

Reviewer #1: Yes

Reviewer #2: Yes

5. Review Comments to the Author

Reviewer #1: - The paper is well written and easy to follow.

There are some minor typos in the text, for example, “different” in the Abstract.

- The main comment is on the presentation of the results. It would be better to have an uniform way of presenting all the results such that they cover the following: state the exact question posed to the respondent (especially important for the reader to know for perception questions), show result in a table with all options percentages and sample sizes. This is essential as sometimes a result is mentioned without the exact statistic. In addition, for each set of outcomes it could be mentioned if no heterogeneity and time varied effects were found.

- Would really want to see the 2x2 table of outcome (health appointment change yes or no) & type of health provider for this result:

"Two-thirds of these changes occurred in private health services (where most of the appointments were planned)."

As if two-thirds of the total appointments are in the private sector, then this conclusion is not really supported:

"Although our convenience sample is likely to be biased and have over- representation of population using private sector providers (either using private health insurance or health subsystems coverage), the magnitude of these changes suggest that private providers did react with rescheduling and cancellation, while not such effect was noticed by the users of the National Health Service. "

- Figure 2: What is “respondent who anticipated groceries”? The term used in Section 2.2.2 “anticipated groceries” is not clear, atleast to me, at all. If the literature uses some other term, then please use that. Else please explain and state the exact question.

- Could add a line at the start on the analysis approach so it is crystal clear for the reader: Repeated cross-sectional dataset that is analysed as a pooled and time series sample. Drop the few (x%) panel observations from the analysis.

- Table 1: change “Answers” to number of respondents or something clearer. Same for valid answers.

- Table 2:

-- Definition of health professional

-- Could present more detailed descriptive stats like count, mean, median, min, max.

-- Full age distribution, If child in household, more education levels

- Was the sampling strategy/outreach the same for the three waves? Mention as would matter for the sample selection.

- Add a line on if the information questions were analysed or not: Search for information, Quality and reliability of information

- Given this is posed as a reserach question, could add the cross-tab of perception categories and behaviour outcomes.

"As such, analysing people’s perception of current events and how that can translate into actions and consumption decisions is of utmost important to understand the effectiveness and collateral damages of the pandemic-related measures."

Reviewer #2: The work is very relevant and contributes directly to our understanding of a pandemic that has gripped the world and affected people from various dimensions.

I have a few points that I urge the authors to address in order to situate the learnings from the paper better.

1) How different is the socio-demographic characterization of the survey from the population distribution (Table 2). A column showing population distribution will be helpful to understand how to extend the learning to the Portugese population. I understand the authors have said this is a limitation, but it would be good to quantify the magnitude of difference in representativeness.

2) I think the authors could use the heterogeneity in survey responses to characterise how successful the lockdown measures implemented were. This will help understand compliance and socio-economic consequences.

3) I urge the authors to correct grammar and spelling errors. Some of the writing can be made more succinct. Finally, I urge them to tighten the writing.

6. PLOS authors have the option to publish the peer review history of their article (what does this mean?). If published, this will include your full peer review and any attached files.

Reviewer #1: No

Reviewer #2: No

---

## [Author Response · Author response to Decision Letter 0]

6 Aug 2020

Dear Professor Renuka Sane,

Thank you very much for the helpful comments and for the opportunity to revise our manuscript. We acknowledge the need of revisions and tried to answer to all yours and reviewers’ observations, as described below. In addition, and given the large number of COVID-19 related papers published in the last months, we have conducted an additional review of the literature and added relevant references to our introduction and discussion. Note that this version also includes the authors list updated order.

We corrected typing errors and extensively rephrased the article, aiming for a better and easier understanding. 

Reviewer #1

Methods – “Was the sampling strategy/outreach the same for the three waves? Mention as would matter for the sample selection.”

o Following the reviewer’s suggestion, we now explicitly state in the Methods section: “The distribution strategy in social media was the same for the three waves. For the third wave we have additionally emailed professional boards, hospitals and patient associations, requesting them to disseminate the survey amongst their associates”.

Methods – “Add a line on if the information questions were analysed or not: Search for information, Quality and reliability of information.”

o To address this comment Information was included in a footnote of the paragraph justifying which questions were analyzed: ”Despite all information collected, we focused our analysis on four main dimensions: social isolation and concern, stockpiling and consumption patterns, health services performance, and economic consequences1. These dimensions were selected based on data completeness, meaningful results and relevance for policy design.

“1. Topics not studied for this paper include: symptoms, search for information, and quality and reliability of information.”

Methods – “Could add a line at the start on the analysis approach so it is crystal clear for the reader: Repeated cross-sectional dataset that is analyzed as a pooled and time series sample. Drop the few (x%) panel observations from the analysis.”

o As suggested by the reviewer, we included in the Methods section: “Our repeated cross-sectional dataset includes the answers of 7,448 respondents, which we analyzed as a pooled and time series sample.”.

Results – “The main comment is on the presentation of the results. It would be better to have an uniform way of presenting all the results such that they cover the following: state the exact question posed to the respondent (especially important for the reader to know for perception questions), show result in a table with all options percentages and sample sizes. This is essential as sometimes a result is mentioned without the exact statistic. In addition, for each set of outcomes it could be mentioned if no heterogeneity and time varied effects were found.”

o This suggestion was addressed in several ways. In order to report the results uniformly we have included in the appendix a table with the exact question and frequencies for each of the dimensions studied. In terms of the heterogeneity of our findings, we have conducted probit analysis that include gender, age, education and region. We have also added a figure to show heterogeneous perceptions about present and future financial status by professional occupation. Variations over time are mentioned in the text.

Results – “Would really want to see the 2x2 table of outcome (health appointment change yes or no) & type of health provider for this result.

o Consistently with the approach for the outcomes in the other dimensions we have studied (bidirectionally) the association between health appointments postponed/cancelled and the place where the appointments were made, adjusting for gender, age, education and region controls (table 6) . Additionally, in table A3 of the appendix we provide de proportion of appointments by private/public and type of provider.

Results – "Two-thirds of these changes occurred in private health services (where most of the appointments were planned)."

As if two-thirds of the total appointments are in the private sector, then this conclusion is not really supported:

"Although our convenience sample is likely to be biased and have over- representation of population using private sector providers (either using private health insurance or health subsystems coverage), the magnitude of these changes suggest that private providers did react with rescheduling and cancellation, while not such effect was noticed by the users of the National Health Service. " 

o We agree with the reviewer and have removed this conclusion, including in the Discussion section only: “A cancellation or postponement seemed to be more probable when scheduled in private health services, contrasting with the public services.”. We have further explored two-way associations for these results with probit models (Table 6) and present the interpretation of the findings: “The regression results indicate that having an appointment scheduled in a private medical service increased the probability of having a cancellation or postponement in13%, significant at 1% level. For public services this probability is close to null and not statistically significant. Gender, age or education were not significantly associated with having an appointment changed.”

Results – “Given this is posed as a research question, could add the cross-tab of perception categories and behaviour outcomes.

"As such, analysing people’s perception of current events and how that can translate into actions and consumption decisions is of utmost important to understand the effectiveness and collateral damages of the pandemic-related measures."”

o In order to address the reviewer’s suggestion, we included in the Results section the analyses of two-way associations between perceptions (i.e. concern) and behavior outcomes (i.e. social isolation and stockpiling) or expectations (i.e. future negative financial impact).

Figures – “Figure 2: What is “respondent who anticipated groceries”? The term used in Section 2.2.2 “anticipated groceries” is not clear, atleast to me, at all. If the literature uses some other term, then please use that. Else please explain and state the exact question.”

o Following the reviewer’s comment, “anticipated groceries” was replaced by stockpiling throughout the manuscript and in the Figure 2.

Tables – “Table 1: change “Answers” to number of respondents or something clearer. Same for valid answers.”

o The details on (valid) answers were provided on Table 1 to make the difference between the number of observations including panel observations (answers) and the cross-sectional aspect of the survey (respondents). As previously recommended by the reviewer panel observations are not used, which we address explain in footnote 6 of page 5. In this context, we no longer present information about answers in Table 1 but only the number of respondents, which correspond to the observations used for this paper.

Tables – “Table 2: -- Definition of health professional 

o Unfortunately we did not provide to the respondents a definition of health professional. We have added a sentence to the note in Table 3 clarifying this: “No definition of health professional was provided in the survey, responses are based on the participants´ understanding of this professional occupation.”

 Tables Could present more detailed descriptive stats like count, mean, median, min, max. 

o To address this comment we have added counts to all the tables in which it applies. Mean, median, min and max do not apply to our descriptive analysis.

Tables Full age distribution, If child in household, more education levels”.

o Following the reviewer comment we have added all the information collected about education and age to Table 2 (age collected as categories and not continuously). Unfortunately, we did not collect data about children in the household.

English proofreading – “There are some minor typos in the text, for example, “different” in the Abstract.”

o As suggested, we extensively reviewed the written English and corrected grammatical and spelling errors.

Reviewer #2

Results – “1) How different is the socio-demographic characterization of the survey from the population distribution (Table 2). A column showing population distribution will be helpful to understand how to extend the learning to the Portuguese population. I understand the authors have said this is a limitation, but it would be good to quantify the magnitude of difference in representativeness.”

o To address the reviewer’s suggestion, we included in Table 2 a column showing the differences between our sample and the Portuguese population demographic and socioeconomic characteristics, for the variables for which comparable information could be retrieved.

Results – “2) I think the authors could use the heterogeneity in survey responses to characterise how successful the lockdown measures implemented were. This will help understand compliance and socio-economic consequences.”

o Following the reviewer’s suggestion we have extended our analysis in terms of heterogeneity, by studying the effect of gender, age and education in the most relevant outcomes (probit regressions and structural equation model). Based on these analyses we are able to describe which individual characteristics were associated with their concerns and behaviours (e.g. social isolation and stockpiling). Given the non-representativeness of our sample, as well as the absence of measures about COVID-cases in our sample, we have opted not to conclude about the success of the lockdown measures based on our results. We can, however, conclude that at least a group of the Portuguese population anticipated isolation measures in relation to the beginning of the partial lockdown in the country, and about the associations between the concerns and the reactions of the Portuguese citizens.

English proofreading – “3) I urge the authors to correct grammar and spelling errors. Some of the writing can be made more succinct. Finally, I urge them to tighten the writing.”

o As suggested, we extensively reviewed the written English and corrected grammatical and spelling errors.

---

## [Editor Report · Decision Letter 1]

25 Aug 2020

PONE-D-20-15289R1

Concerns and Adjustments: How the Portuguese population met COVID-19

PLOS ONE

Dear Dr. Valente de Almeida,

Thank you for submitting your manuscript to PLOS ONE. After careful consideration, we feel that it has merit but does not fully meet PLOS ONE’s publication criteria as it currently stands. Therefore, we invite you to submit a revised version of the manuscript that addresses the points raised during the review process.

There are only two requirements:

a. Can you please add the caveat that this is not a representative sample in your abstract? (or where you discuss the results in brief on the first page). This is important as it states the shortcomings of this research upfront.

b. Check for spelling errors again. For example, I found that you use "depende" instead of "depends" (perhaps)? Please do a proof-reading once again.

We look forward to receiving your revised manuscript.

Kind regards,

Renuka Sane

Academic Editor

PLOS ONE

---

## [Author Response · Author response to Decision Letter 1]

17 Sep 2020

Dear Editors, 

We appreciate the feedback provided. Please find attached the revised version. According to your feedback we have carefully revised the wording/ phrasing of the document, as to improve its style. Additionally, we have included a specific mention regarding the non-representative sample both in the abstract and results section – in addition to the references made in the Discussion section.

Kind regards,

Sara Valente de Almeida

Eduardo Costa

Francisca Vargas Lopes

João Vasco Santos

Pedro Pita Barros

---

## [Editor Report · Decision Letter 2]

29 Sep 2020

Concerns and Adjustments: How the Portuguese population met COVID-19

PONE-D-20-15289R2

Dear Dr. Valente de Almeida,

We’re pleased to inform you that your manuscript has been judged scientifically suitable for publication and will be formally accepted for publication once it meets all outstanding technical requirements.

Kind regards,

Renuka Sane

Academic Editor

PLOS ONE
---

## [Editor Report · Acceptance letter]

2 Oct 2020

PONE-D-20-15289R2 

Concerns and Adjustments: How the Portuguese population met COVID-19 

Dear Dr. Valente de Almeida:

I'm pleased to inform you that your manuscript has been deemed suitable for publication in PLOS ONE. Congratulations! Your manuscript is now with our production department. 

Kind regards, 

on behalf of

Dr. Renuka Sane 

Academic Editor

PLOS ONE